# Social grooming efficiency and techniques are influenced by manual impairment in free-ranging Japanese macaques (*Macaca fuscata*)

**Jenny Paola Espitia-Contreras**[ID][1]*, **Linda M. Fedigan**[2], **Sarah E. Turner**[1]

**1** Department of Geography, Planning and Environment, Concordia University, Montreal, Quebec, Canada,
**2** Department of Anthropology and Archaeology, University of Calgary, Calgary, Alberta, Canada

\* jepaesco@gmail.com

**Data Availability Statement:** The data underlying the results presented in the study are available from Scholars Portal Dataverse, the Concordia University Open Access repository, at DOI: https://doi.org/10.5683/SP2/9DRWP5.

## Abstract

Animals born with physical impairments may particularly require behavioural flexibility and innovation to survive and carry out social activities, such as grooming. Studies on free-ranging Japanese macaques on Awaji Island, Japan, have shown that individuals with congenital limb malformations exhibited compensatory behaviours while grooming, such as increased mouth and elbow use for removing ectoparasites. The aim of this study is to explore disabled and nondisabled grooming techniques to determine whether and to what extent disabled monkeys develop novel grooming techniques, and if there is disability-associated variation in grooming efficiency. We hypothesized that modified grooming techniques used by disabled monkeys fulfilled the social and relaxing functions of grooming, however, that grooming by manually impaired individuals may still carry a hygienic cost to the recipients. Grooming behavioural data were collected by video in 2007 on 27 adult females (11 with CLMs). With a detailed grooming-related ethogram, we transcribed 216 2-minute continuous grooming video samples. We analyzed the data using generalized linear mixed effects models in R. We found that monkeys with manual impairment were less efficient groomers, as measured by removal and movement efficiency during grooming. However, there were no significant differences associated with the number of grooming movements per sample among the focal animals. Additionally, with a behavioural sequential analysis, we isolated 8 distinct grooming techniques and 3 novel disability-specific movements. Our results indicate that innovation and modification of movement types does not entirely compensate for manual disability, and that manual impairment carries a cost to the hygienic function of grooming. However, for the grooming recipient, the experience of being groomed by a disabled or nondisabled groomer is likely similar, and through movement compensation, disabled monkeys are able to engage in the social aspect of grooming without incurring any disability-associated costs.

## Introduction

Free-ranging and wild animals are exposed to many environmental and social challenges, changes and conditions that can lead them to modify their behaviours or innovate novel behaviours in order to survive and reproduce [1]. Such behavioral flexibility or behavioural phenotypic plasticity can lead to innovative behaviours. Higher incidences of innovative

**Funding:** SET gratefully acknowledges the funding provided by the Fonds de Récherche de Québec—Nature et Technologies (FRQNT Grant #25467) and PhD research funding from The Leakey Foundation, an Izaak Walton Killam Memorial Scholarship, the Animal Behavior Society, Province of Alberta Graduate Scholarships, the University of Calgary, the University of Calgary Dept. of Anthropology and NSERC (post-graduate scholarship B). LMF is grateful for funding from NSERC and the Canada Research Chairs Programme. The funders had no role in study design, data collection and analysis, decision to publish, or preparation of the manuscript.

**Competing interests:** The authors have declared that no competing interests exist.

behaviours have sometimes been found to occur among individuals that are considered to be at a competitive disadvantage, including females in sexually dimorphic species where males are larger than females, juveniles, and low-ranking individuals [1–3]. Physically impaired individuals may also show behavioural flexibility as a response to lower competitive advantage and as compensation for disability [4–6]. Physical impairment and long-term physical disability can influence an animal's fitness, or fitness proxies (e.g., reflected in difficulties finding potential mates or social partners, and reduced locomotor abilities that can increase risk of injury, early mortality, and predation) [2,6–9].

However, animals are often able to compensate behaviourally for physical impairment. Examples of compensatory behavioural responses were observed in fruit flies that, after leg-amputation, recovered their unbiased walking [10]. Ang and colleagues (2016) [11] observed a captive chimpanzee who preserved full mobility of her shoulder after a forelimb amputation, and was able to choose paths and climbing techniques that compensated for her physical impairment. Furthermore, snare-injured wild chimpanzees and gorillas modified feeding techniques according to their individual needs, which varied according to the degree of impairment [4,5].

A group of Japanese macaques with a high instance of individuals born with limb malformations, presents an opportunity to examine the influence of congenital impairments on the behaviour of free-ranging animals. The cause of these malformations of the limbs and digits remains unclear; however, the population is small and isolated, and the presence of pesticide residues in provisioned food has been suggested as a contributing factor involved in the occurrence of such malformations [12,13]. In studies conducted on the Japanese macaque population at Awajishima Monkey Center in Japan, in which 17.1% of the censused individuals were affected by CLMs in 2004, researchers found that disabled individuals modified behaviours (e.g., feeding, scratching, and grooming) to compensate for their impairments, and generally received undifferentiated social treatment from conspecifics [6,14,15].

Social grooming or allogrooming (hereafter grooming) in non-human primates and other animals has a hygienic function, lowering ectoparasite burdens, as well as a social function associated with the maintenance of social group cohesion [16–18]. Additionally, studies have shown that after giving and receiving grooming, individuals experienced a reduction in anxiety levels and lower heart rates [17]. Other benefits of grooming are short-term distress reduction, production of β-endorphins [19], and increased tolerance levels [20]. Grooming has been conceptualized as a commodity in a biological market that is traded among animals in exchange for benefits [21]. Empirically it has been found that grooming was traded for agonistic support in adult chimpanzees [22] and female Japanese macaques [23,24]; and for acceptance as a potential mate in long-tailed macaques [25]; and for grooming or huddling in Japanese macaques [17,26].

In nondisabled Japanese macaques, louse egg handling techniques that occur during grooming (i.e., sequences of movements used and efficiency of egg removal) have been found to vary according to maternal kinship [27]. These kin-related variations in technique suggest that some grooming behaviours are socially transmitted. Moreover, movement variations exhibited a rapid diffusion among matrilines, implying imitative learning by observation and trial and error [27,28]. Studies on behavioural flexibility in Japanese macaques at AMC have shown that monkeys with manual impairments used at least some disability-specific movements while grooming. For instance, some individuals with manual impairment used their mouth directly to remove louse eggs or used other variants in grooming, such as using the elbow while searching for eggs [6,14]. However, grooming involves a more complex series of movements and behaviours than was detailed in these studies, and the specifics of disabled and nondisabled grooming techniques were not described or quantified in detail. Furthermore, the

relative efficiency of these techniques and their usage frequency by Japanese macaques with and without physical impairments remains to be investigated. Examining the details of grooming sequences can inform our understanding of the specific ways in which disabled individuals manage compensation for their impairments, through modification of commonly-used grooming techniques or the innovation of novel movement patterns. Furthermore, such investigations can also improve our understanding of the costs associated with these physical impairments.

In this study, we address the following research questions: 1) Are primates with extensive manual impairments able to compensate for their disabilities during allogrooming, and if so, how do they compensate? 2) Are disabled female Japanese macaques able to fulfill the hygienic function of grooming, assessed as grooming efficiency (comprised of removal efficiency: number of eggs removed per sample, and movement efficiency: number of movements per egg removed)? 3) Are disabled female Japanese macaques able to fulfill the social and relaxing functions of grooming, as measured through frequency of grooming gestures used? 4) Do disabled females employ novel grooming techniques and/or innovate novel grooming gestures? 5) Is there a difference in the number and variety of grooming techniques used by disabled and nondisabled females?

We hypothesized that disabled females would show significant differences in their grooming behaviours in comparison to nondisabled females, and we expected that disability would carry a cost in terms of manual dexterity and would affect the grooming techniques used by disabled individuals, as well as their grooming efficiency. However, we also hypothesized that manually impaired females would be able to compensate behaviourally for their physical impairments. In particular, while we expected that the movements developed by disabled individuals would not meet the hygienic efficiency of nondisabled females, we expected that they would use a similar gesture rate in grooming, which would allow them to fulfill the social function of grooming. Evidence of behavioural flexibility could emerge from our results in a number of ways: a finding of no significant difference in grooming efficiency, with substantial variation in grooming techniques could suggest that behavioural flexibility compensates for disability caused by CLMs; individuals could also demonstrate behavioural flexibility through the innovation and use of novel grooming techniques, or through modification of commonly used grooming gestures and sequences. In any case, we would expect to see a wider variation in grooming techniques among disabled compared to nondisabled individuals.

## Methods

All data collection was non-invasive and in accordance with the requirements of the University of Calgary Animal Care Committee (Protocol number: BIO8R-03) and those of the Awajishima Monkey Center in Japan, with video data collection protocols that have also been approved by the Animal Use Committee at Concordia University (Protocol number 30009663).

### Study site and subjects

The Awajishima Monkey Center (AMC), located on Awaji Island in the Seto Inland Sea of Japan, is a privately-owned tourist destination and field research station with a free-ranging and provision-fed population of ~400 Japanese Macaques [15]. Physical impairment in the form of congenital malformations of the limbs and digits (CLMs) is quite common in this population (17.1% of the individuals had CLMs) [15]. Study subjects included 27 adult females, of which 11 had CLMs and 16 were nondisabled controls (Table A in S1 File). We only examined adult female grooming, in order to control for sex and life-stage variability, and because

specific grooming behaviours (and styles of grooming, defined as consistent, detailed sequences of grooming behaviour) has been found to be transmitted among female kin [27,28]. The subjects with CLMs varied in terms of their impairment level. To account for and quantify this variation in the form and extent of individual physical impairments, the subjects were assessed according to an index of manual disability (Table A in S1 File). This 0–1 index was modified from a general index of disability created for this population [15]. Each individual's forelimbs were visually assessed for absence and malformation of fingers, thumbs and forearms, with absence weighted more heavily than malformation, and thumbs weighted more than fingers (S. Turner, unpublished data).

## Video and data collection

The videos were recorded at the AMC from May to July 2007, using a Sony camcorder. Videos were collected with the goal of obtaining continuous focal animal data [29]. Each focal subject was filmed from a distance of between 1 to 5 meters while they were engaged in grooming a partner. Sampling order was determined using a semi-random protocol, with efforts made to collect the same number of samples per subject; we drew a random order, and attempted to locate the next grooming individual in the random order; however if we could not locate a particular individual, we would move to the next available individual, prioritized according to the random order.

## Data entry

To analyze the videos, we created a grooming behavioral ethogram including movements performed by disabled and nondisabled Japanese macaques (Table B in S1 File). For the focal-video analysis, we used Behavioural Observation Research Interactive Software (BORIS) [30]. Behavioural data from the video were recorded with a slowed playback speed of 0.2 times original speed. We examined the data and determined that 2-minute sample length maximized the use of the available data, while not compromising the independence of the samples. Our sampling protocol included the following criteria: 1) behaviours not related to grooming such as scratching, auto grooming, travel, or handling infants could not exceed 13 seconds; 2) samples from the same recording session had a minimum of 2 minutes separation; 3) each focal sample had only one grooming recipient.

As louse eggs are very small (~1–2 mm), it was not always possible to see the egg itself in the video (or by direct observation of the focal subject). However, since removing a louse egg involves the use of fine motor skills (gripping the egg and doing a pinch) or precision use of the mouth, individuals often required several attempts to successfully complete this action. For these reasons, we defined the following criteria for identifying a successful louse egg removal when the egg itself was not visible on the video: when the subject used her hand to detach the egg and bring it to her mouth, or used her teeth directly to detach the egg, the gesture had to be followed by a change in the groomed area to be counted; if the subject licked the area instead of using her teeth, this was not counted as an egg removal.

## Statistical analysis

**Grooming efficiency.** A total of 216 2-minute video focal grooming samples were analyzed. For assessing the grooming efficiency of the subjects, we recorded the number of grooming movements and the number of eggs removed per sample. We calculated the average number of movements performed per egg removed and determined how often the mouth was used to remove an egg. Infant-related behaviours and other behaviours were not included in

the analysis (Table B in S1 File). Grooming efficiency was measured by two variables: removal efficiency (the number of eggs removed per sample), and movement efficiency (the number of movements per egg removed).

We used Generalized Linear Mixed-Effects Models (glmer function in R) to analyze the quantitative grooming data [31]. Mixed effects models allow for the inclusion of random factors, which addressed the problem of pseudo-replication associated with having multiple samples on each individual monkey and minimized the influence of having different numbers of samples per individual [32]. In each model, we verified that data met the required assumptions for parametric statistics: that the residuals were normally-distributed, and that variance was homogeneous. We visually assessed the error distribution in Q-Q plots and tested normalcy using Shapiro tests [33,34].

We ran three sets of models to test our hypotheses. We ran each set of models with manual physical impairment as the fixed factor and individual focal monkey as a random factor. Each model was run using a categorical measure of disability (disabled or nondisabled) and a continuous measure of manual disability (index of manual disability) as fixed factors (Table C in S1 File). In addition, we considered age and matriline (kinship) as potential additional predictor variables, but did not find any significant relationships, and therefore did not retain them in the final mixed effects model analysis (Table C in S1 File). In model 1a and 1b, we tested the effect of manual physical impairment on louse egg removal efficiency. In model 2a and 2b, we tested the effect of manual physical impairment on the number of movements performed in 2 minutes. In model 3a and 3b, we tested the effect of manual physical impairment on movement efficiency (samples where no eggs were removed were excluded). For this last set of tests, we log-transformed the data, and used a generalized linear mixed model with a Penalized Quasi-Likelihood and an Inverse Gaussian distribution to perform this analysis. Furthermore, we compared the frequency of use of the mouth while grooming between disabled and nondisabled females using a non-parametric Mann-Whitney U-test, as these data did not meet the requirements for parametric analysis.

**Grooming technique flowcharts.** We created grooming behavioural flowcharts for all disabled and nondisabled subjects to visualize the sequence of grooming movements used and to define and examine variation in grooming styles. We modeled these flowcharts on the Transitional Probabilities Matrices from a study on foraging lizards [35]. These matrices indicate the probability of transitioning from any given (grooming) movement to another. We generated individual probability matrices from the data using Joint Frequencies Matrices with the Generalized Sequential Querier GSEQ over all the recorded movements of each subject, excluding any non-grooming related behaviours [36].

Using DOT, a graph description language, we designed a flowchart for each grooming technique including all the performed movements and the transitions among them. Transitions were only included in the chart if their probability was higher than 15%. That is, a transition between movements X and Y was drawn if either the probability of doing movement X before movement Y (lag -1) or the probability of doing Y after X (lag 1) was higher than 15% [36]. On the flowcharts, thicker lines represent high probability transitions. To distinguish high transition probability, the transition probabilities were summed in matrices from before (lag -1) and after (lag 1). Transitions with sum higher than the mean, were defined as having high probability (See Matrices and transition probabilities in S3 File).

We evaluated the flowcharts qualitatively to identify grooming styles and the use of disability-specific techniques. Finally, we used Mann-Whitney U-tests to determine if grooming efficiency varied significantly among grooming styles. For all statistical tests in this study, the results were considered to be statistically significant when P values were < 0.05.

## Results

Full statistical results are available in S2 File. The total number of observed eggs removed in the study was 837 and the total number of movements included in the analysis was 30,798.

### Effects of disability on grooming efficiency

Removal efficiency of nondisabled females was significantly higher than their disabled counterparts: on average, nondisabled females removed 71.45% more eggs than disabled females (Poisson GLMM: Estimate = 0.80, Std. Error = 0.21, z-value = 3.79, p-value <0.001). Likewise, the extensiveness of manual impairment (as measured by the index of manual disability) also had a significant negative effect on removal efficiency; as the index value increased, the number of eggs removed decreased (Poisson GLMM: Estimate = -0.60, Std. Error = 0.09, z value = -6.77, p-value <0.001) (Fig 1).

There was no significant disability-associated difference in the number of grooming movements performed per 2-minute sample (disability category, Poisson GLMM: Estimate = -0.04, Std. Error = 0.05, z value = -0.83, p-value = 0.41; Index of disability, Poisson GLMM: Estimate = -0.05, Std. Error = 0.085, z value = -0.578, p-value = 0.56).

Movement efficiency was significantly and negatively associated with disability (Inverse Gaussian GLMM: t-value = 3.76, df = 25, p-value < 0.001) and the index of manual disability (Inverse Gaussian GLMM: t-value = -4.42, df = 25, p-value < 0.001). The results reveal that disabled females used more movements per egg removed than nondisabled conspecifics and that as the index of disability increased, the number of movements per egg removed increased as well (Fig 2).

Disabled females used their mouth to groom significantly more frequently than their nondisabled counterparts (Mann-Whitney: W = 9879.5, p-value < 0.001) (Fig 3).

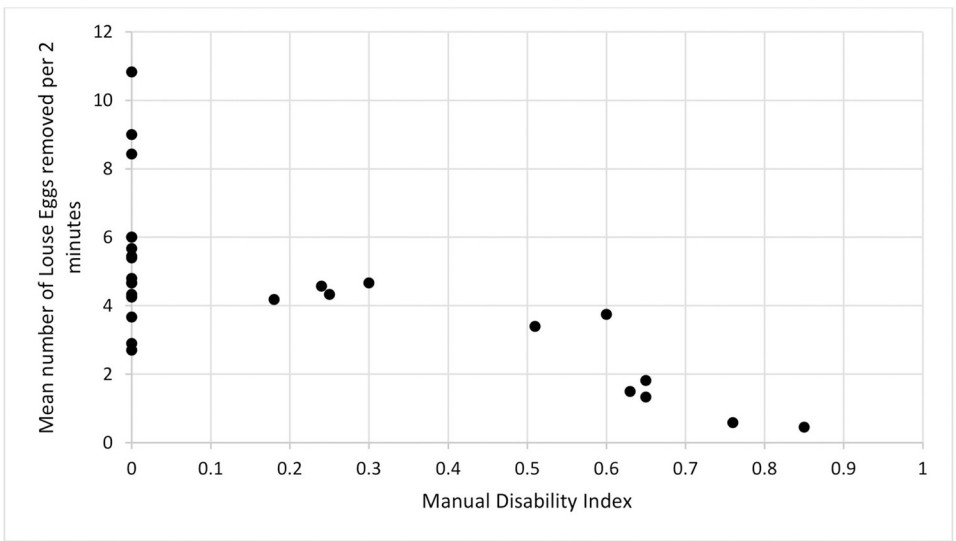

**Fig 1. Mean number of louse eggs removed per 2-minute sample as a function of manual disability index.** Each black dot on the chart represents the mean for one individual monkey. The black dots on the vertical axis each represent a nondisabled individual's mean number of louse eggs removed per 2 minutes. The dots dispersed along the horizontal axis represent each individual with a degree of manual impairment measured using an index of disability from 0 to 1, where 0 represents a nondisabled monkey and 1 represents the complete absence of forelimbs. Group means: Nondisabled females (5.26 eggs/sample) Disabled females (2.49 eggs/sample).

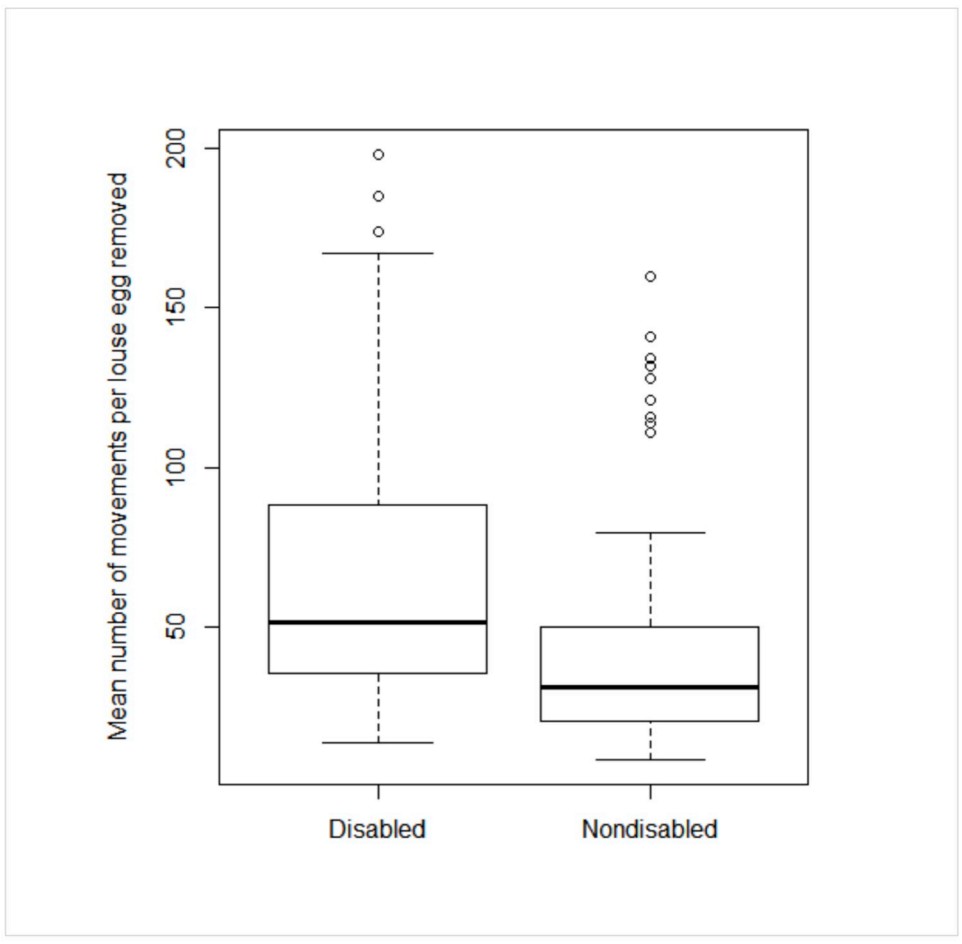

**Fig 2. Mean number of movements per louse egg removed for disabled and nondisabled individuals.** This plot illustrates the median values for the mean number of movements per louse egg removed for disabled and nondisabled monkeys (thick lines inside the boxes). The circles represent the outliers and the dashed lines show the range. The data variability is shown in the top and bottom lines of the boxes, representing the inter-quartile range (25th and 75th percentiles). The whiskers and circles show the maximum and minimum values of the samples. Group means: Nondisabled females (42.23 movements/egg removed) Disabled females (64.71 movements/egg removed).

## Grooming techniques

We found that female Japanese macaques groomed in a predictable sequence, consisting of four stages performed in the following order: 1) Find the egg, which comprised all movements used for parting the hair and locating the egg; 2) Grip the egg, with the movements to loose, grip, and see the louse egg after its removal; 3) Carry egg to mouth, including the movements to bring the louse egg to the mouth; and 4) Eat removed egg, with a mastication movement (Fig 4). Based on the variations in movement use, movement type, and transition frequencies, as well as significant differences in removal and movement efficiency, we identified the existence of two nondisabled and six disabled distinct grooming techniques.

**Nondisabled techniques.** Some of the nondisabled females (6/16) were observed to use the mouth directly for removing louse eggs in 13 out of 15,119 transition frequencies. As a result, we identified two distinct nondisabled grooming techniques associated with movement variation in the third stage: 1) the main nondisabled grooming technique (ND) and 2) the modified nondisabled grooming technique (NDM). Individuals that used the *Mouth directly*

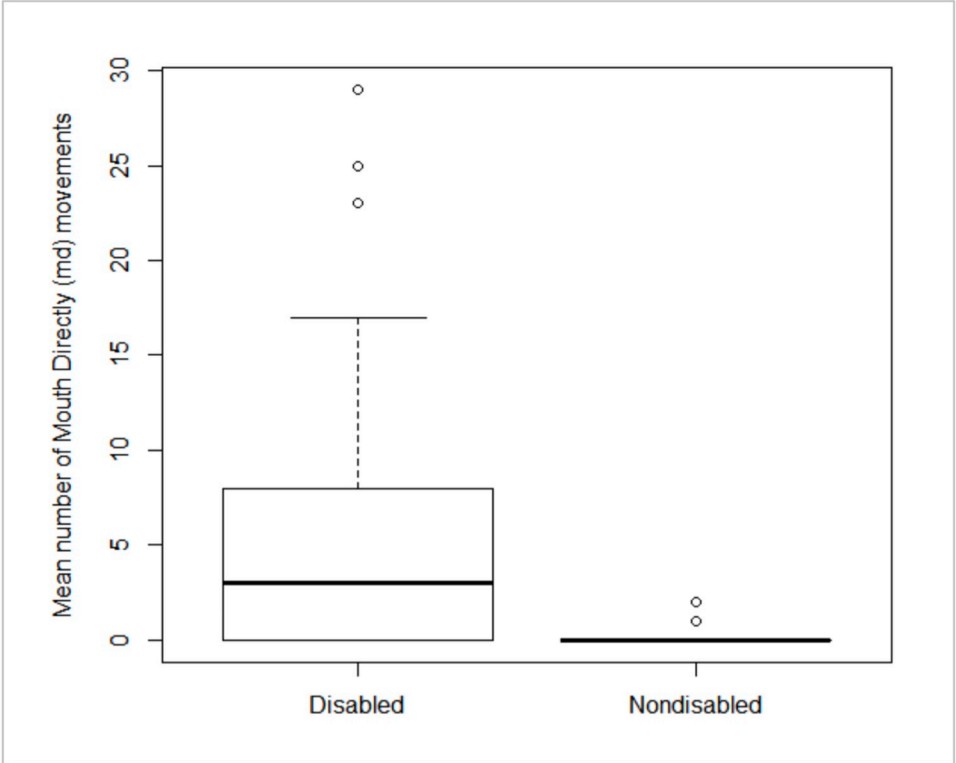

**Fig 3. Mean Number of mouth directly (md) movements for disabled and nondisabled individuals.** This plot illustrates the median values for the mean number of Mouth directly movements removed for disabled and nondisabled monkeys (thick lines inside the boxes). The circles represent the outliers and the dashed lines show the standard deviation. The top and bottom lines of the boxes represent the inter-quartile range (25th and 75th percentiles). The whiskers and circles show the maximum and minimum values of the samples. Group means: Nondisabled females (0.13 md movements), Disabled females (4.78 md movements).

(md) gesture were 17.58% less movement efficient than those who did not (Mann-Whitney: W = 1098.5, p-value = 0.042). However, the removal efficiency was not significantly different between both groups (Mann-Whitney: W = 1680, p-value = 0.155) (S4 File).

In first nondisabled technique (ND) (Fig 5), the individual starts with the *Find the egg stage* by using a *Hand/arm push* (hp) movement for parting the hair with one hand followed by a *Finger sweep* (fs) movement through the hair with the index finger of the opposite hand, to look for louse eggs. Then, she uses the *Two digit pull* (sp) movement to make the skin area more visible. While in this stage, she may use the *Grab groomee's limb* (gl) movement to accommodate the groomee and inspect a different area. She will continue in this stage until she finds an egg to remove. Once an egg is located, the *Grip the egg stage* starts, she can use her nail for loosening the adhesive of the louse egg with a *Thumb/nail loosening* (nl) gesture or directly use her index and thumb to do a *Two-digit pinch* (dp) to remove the egg. Then, to confirm that the egg was removed from the hair, she checks her hand with a *See egg on finger/hand* (se) movement. If she observes that the egg is still attached to a pulled hair, she uses *Second-hand support* (sh) for detaching the egg and checks again if the egg is completely hair-free. Next, in the *Carry egg to mouth* stage, she brings the egg to her mouth with the hand holding the egg using the *Single hand to mouth* (sm) movement. Sometimes, she may look at her hand again to make sure the egg is no longer in her hand. Finally, at the *Eat removed egg* stage, she does a *Louse egg mastication* (em) gesture for eating the egg. After this, she goes back to the

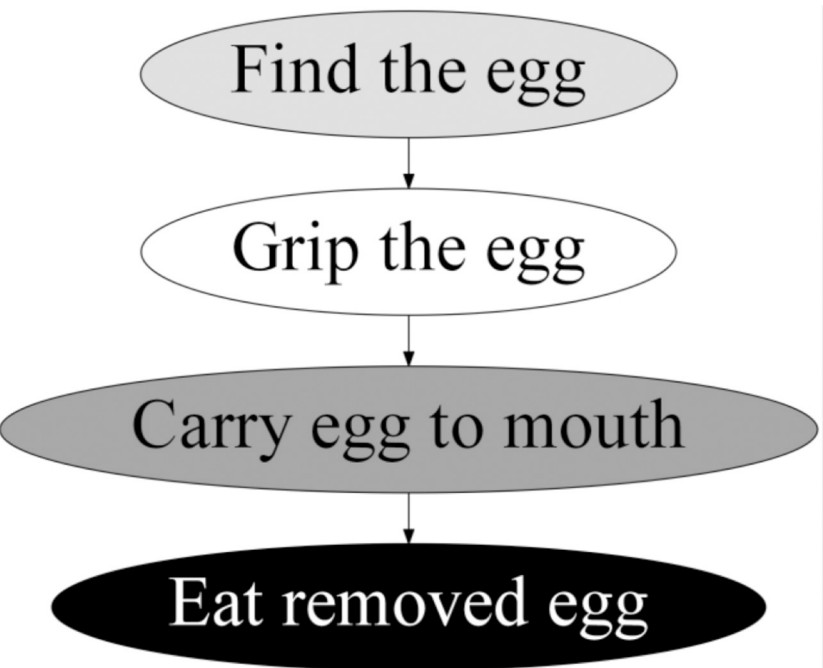

**Fig 4. Grooming stages.** The process begins at the top of the figure. These grooming stages were observed in all subjects during the study.

first stage either with a *Finger sweep* (fs) or a *Two-digit pull* (sp) if she continues grooming in the same area, or with a *Hand/Arm push* (hp) to continue somewhere else.

The second nondisabled technique is a modification of the main technique (NDM) (Figure B in S3 File). The NDM technique follows the same sequence of movements overall but includes the use the *Mouth directly* (md) movement in the *Carry egg to mouth* stage. After locating an egg, they may skip the *Grip the egg* stage and remove it directly with their mouths. The transitions going towards the *Mouth directly* (md) movement are dotted because its frequency was lower than 15%. The use of *Nail loosening* (nl) (10/16), *Grab groomee's limb* (gl) (15/16), and the *Second-hand support* (sh) (14/16) movements were not observed in all nondisabled individuals, however the associated transition variations did not meet the criteria for defining a separate grooming technique.

**Disabled techniques.** The grooming techniques of disabled females follow the same stages described for nondisabled females, but they introduced novel movements to compensate for their physical impairments. Depending on the degree of manual disability, these individuals adapted nondisabled movements, using their impaired upper limbs for grooming or using the mouth more frequently for removing louse eggs. We identified 6 different grooming techniques and they are presented in order of similarity to nondisabled techniques, starting with the ones that share more similarities in movements and transitions with the main nondisabled grooming technique (ND). Additionally, the *Mouth directly* (md) movement was common among all disabled groomers, and we classified it as disabled-associated gesture; although this gesture is used occasionally by nondisabled individuals, disabled monkeys used it significantly more than the nondisabled individuals (See Fig 3).

The first disabled technique DA (Fig 6) was used by the four individuals with the lowest manual index of disability scores in the sample (Index of disability = 0.18 (Yokam), 0.24

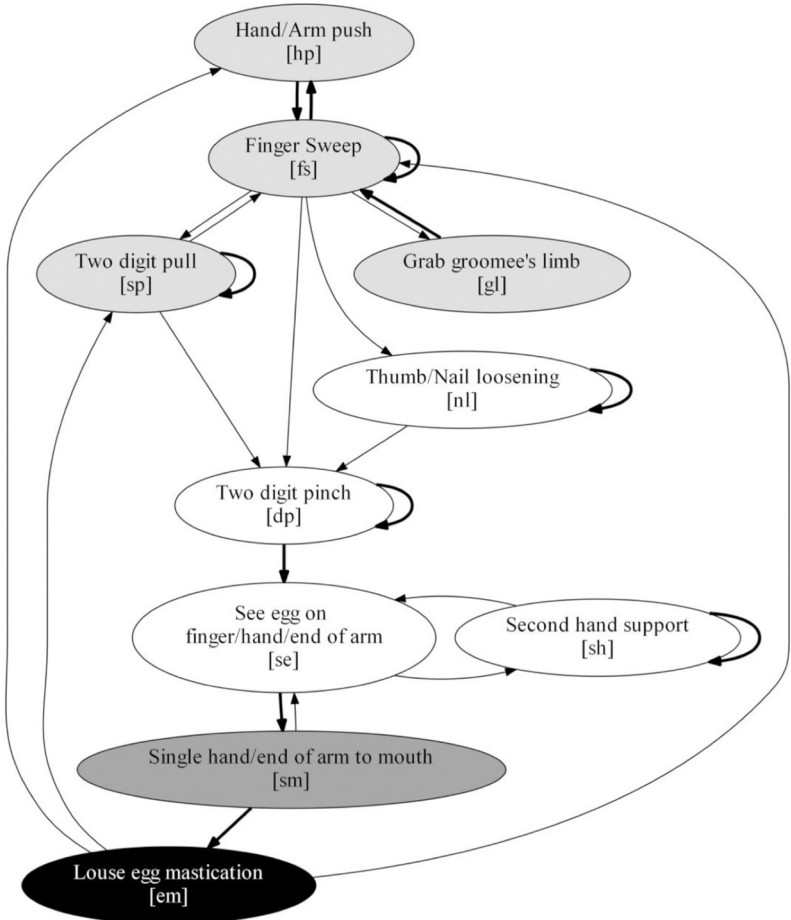

**Fig 5. Main nondisabled grooming technique (ND).** The process begins at the top of the figure. Node color matches with the four grooming stages described in Fig 4. Lines represent relevant transitions. Thicker lines represent transitions with probabilities higher than 50%.

(Nachan), 0.25 (Wendy), and 0.30 (Ran)). All of these females' hands had some digits absent or malformed but retained their ability to use a thumb-finger pinch grip [14]. In their grooming process, these individuals used the same movements used in the nondisabled techniques, including the use of the mouth. However, their technique also included transitions not found in the main nondisabled technique (ND), such as doing a *Hand/Arm push* (hp) followed by *Grab groomee's limb* (gl). This technique showed significantly higher movement efficiency than the ND technique (Mann-Whitney: W = 1240, p-value = 0.012), but was not significantly different from the NDM technique (Mann-Whitney: W = 879, p-value = 0.619). There was no significant difference in the removal efficiency when compared with the ND technique (Mann-Whitney: W = 824.5, p-value = 0.272), nor when compared with the NDM technique (Mann-Whitney: W = 861, p-value = 0.739) (Table C in S4 File).

The second disabled technique DB (Figure D in S3 File) was used by two individuals with medium manual disability index scores (0.51 (Kinchan), 0.6 (Fumin)). They both had some digits absent on both hands and limited pinching ability [14]. They used the same movements as the nondisabled technique and they also included the *Two hand/arm to mouth* (tm) movement in the *Carry egg to mouth* stage. In this technique, after gripping the egg, the groomer either used their mouth to remove the egg, brought a single hand to the mouth or, in the most

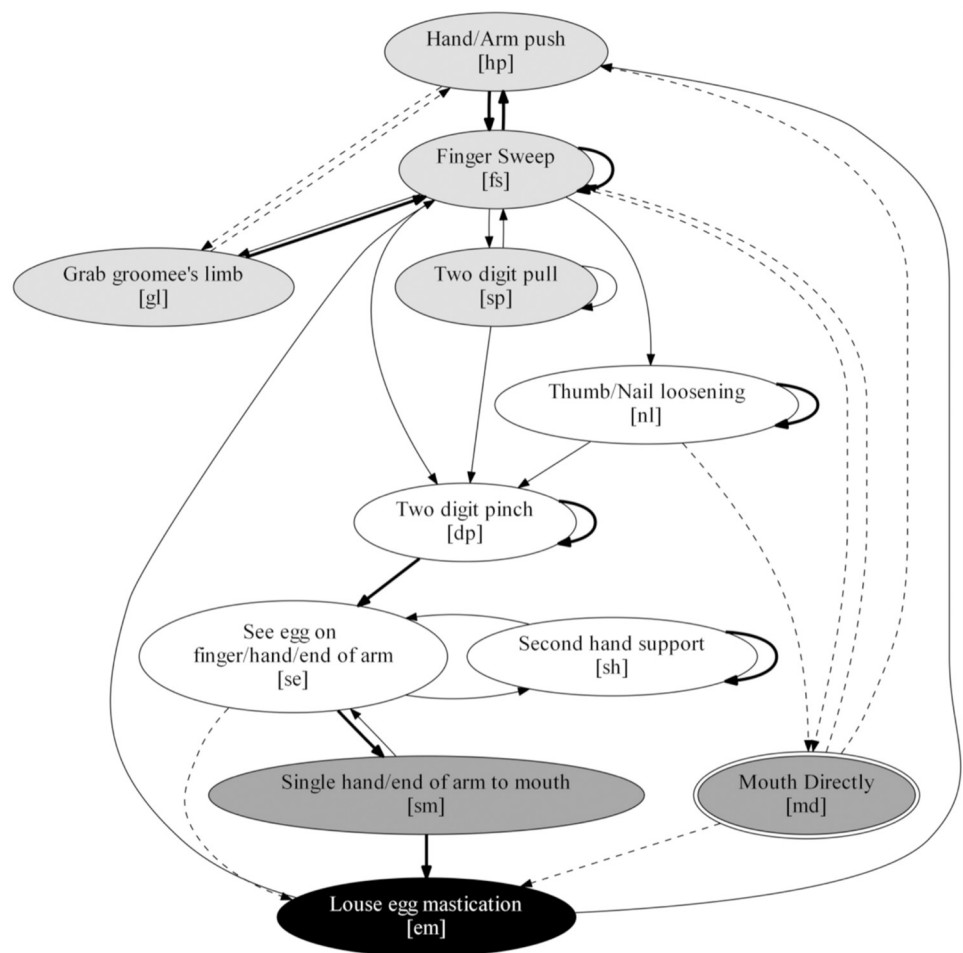

**Fig 6. Disabled grooming technique DA.** The process begins at the top of the figure. Node color matches with the four grooming stages described in Fig 4. Solid lines represent transitions common with the ND technique. Dashed lines show transitions not found in the ND technique. Transitions with probabilities higher than 57% are indicated with thicker lines. Double-bordered movements (1) are performed only by the disabled individuals that use this technique.

frequent transition, they brought both hands or arms directly to the mouth, followed by the *See egg on hand/ end of hand* (se) movement, and finishing with a *Louse egg mastication* (em) movement. With this disabled technique DB movement efficiency was significantly lower than for nondisabled technique ND (Mann-Whitney: W = 498, p-value = 0.014), but it was as efficient as the NDM technique (Mann-Whitney: W = 381.5, p-value = 0.149). No difference in the removal efficiency was found when compared with the ND technique (Mann-Whitney: W = 259, p-value = 0.079), nor when compared with the NDM technique (Mann-Whitney: W = 275.5, p-value = 0.401). (Table C in S4 File).

Two individuals with medium-high manual disability index (0.63 (Kobato) and 0.65 (Punch 98)) use the third disabled technique DC (Figure E in S3 File). One of them had one hand absent and the other hand malformed with limited use of her fingers, and the other macaque's hands were both almost entirely absent, each hand having only one malformed finger [14]. Due to their physical impairments, some of the nondisabled movements were not present at the *Find egg* and *Grip egg* stages. In the latter stage, due to their lack of opposite

thumbs or fingers, they were unable to do the *Two-digit pinch* (dp) movement and instead used the distil end of both arms for gripping the egg with a new gesture: *Two hand/arm pinch* (tp). Then, after looking at the egg on their hands, they frequently used *Two hand/arm to mouth* (tm), another new movement, at the *Carry egg to mouth* stage. In the same stage and with a similar frequency, they used the *Mouth directly* (md) gesture to remove and eat the egg. Also, although less frequently, they sometimes used a single hand to carry the egg to the mouth. The individuals using this technique had a significantly lower movement and removal efficiency compared to individuals using the ND technique (movement efficiency: Mann-Whitney: W = 969.5 p-value <0.001; removal efficiency: Mann-Whitney: W = 220, p-value <0.001) and the NDM technique (movement efficiency: Mann-Whitney: W = 785, p-value < 0.001; removal efficiency: Mann-Whitney: W = 232, p-value <0.001) (Table C in S4 File).

A single female used the fourth technique DD (Figure F in S3 File) and had the same medium-high manual disability index (0.65 Pikaru) as one of the females using technique DC, but her distinct physical impairment limited her grooming abilities, and as consequence she used a unique grooming technique. She had both hands malformed, and only one digit on each forelimb [14]. In this technique, the movements that require the thumb and index finger for pulling the hair or doing a pinch to grip the egg were not present. Instead, in the *Grip egg* and *Carry egg to mouth* stages, the individual used the *Mouth directly* (md) movement to remove the louse egg and eat it. The individual using this technique had a significantly lower movement and removal efficiency compared to individuals using the ND technique (movement efficiency: Mann-Whitney: W = 362, p-value <0.01; removal efficiency: Mann-Whitney: W = 80, p-value <0.001) and the NDM technique (movement efficiency: Mann-Whitney: W = 291, p-value = 0.041; removal efficiency: Mann-Whitney: W = 82, p-value < 0.001) (Table C in S4 File).

The fifth disabled technique DE (Figure G in S3 File) was used by a female with high manual disability index (0.76 Yuki). She has no pinching ability since both hands were absent with only a single non-functional digit on each [14]. One frequently used path in this technique started with a *Hand/arm push* (hp) followed by the modified *Hand/arm sweep* (hs) in the *Find the egg* stage, and a *Mouth directly* (md) gesture to remove the egg. However, this sequence did not end with a *Louse egg mastication* (em) gesture because the female never followed the rule of changing the grooming area after using her mouth. In contrast, when using the new *Two hand/arm pinch* (tp) movement, followed by the novel *Two hand/arm to mouth* (tm), the individual checked if the removed egg was on her hand and then did the *Louse egg mastication* (em) movement, while changing the grooming area that confirmed the egg removal. In this technique, only 5 transitions were shared in common with the main nondisabled technique (ND). This technique had a significantly lower movement and removal efficiency compared to the ND technique (movement efficiency: Mann-Whitney: W = 246, p-value <0.01; removal efficiency: Mann-Whitney: W = 33.5, p-value < 0.001) and the NDM technique (movement efficiency: Mann-Whitney: W = 207, p-value = 0.017; removal efficiency: Mann-Whitney: W = 28, p-value < 0.001) (Table C in S4 File).

Finally, the sixth disabled technique DF (Fig 7) was used by the female with the highest manual disability index (0.85, Ribbon). She had both hands completely absent with short forearms and limited elbow flexibility [14]. While grooming, she followed the same order of movements as the nondisabled individuals, however she completely skipped the *Grip egg* stage. Due to the lack of hands, she used a novel *Elbow push* (ep) movement for parting the hair to find louse eggs. With the opposite arm, she performed a modified *Hand/arm sweep* (hs) to look for eggs in the groomee's hair. Then, she always used the *Mouth directly* (md) gesture to remove the egg, followed by a *Louse egg mastication* (em) movement to eat the egg. This is the least

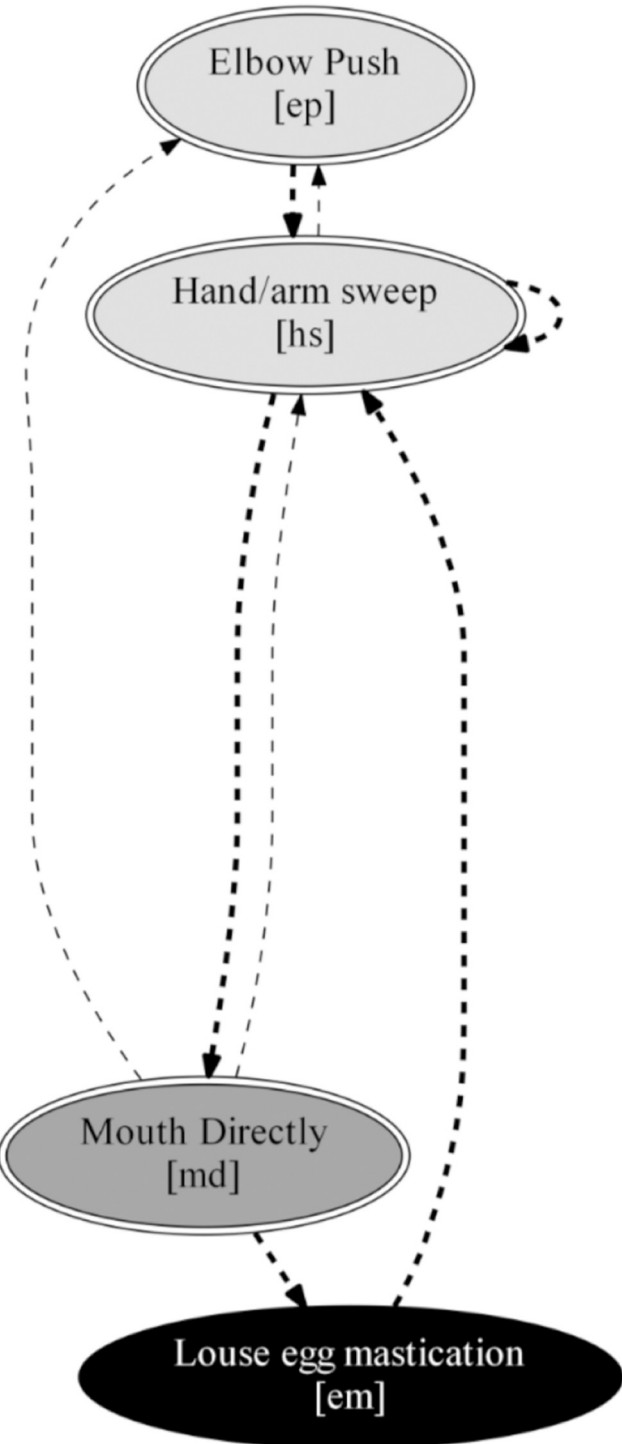

**Fig 7. Disabled grooming technique DF.** The process begins at the top of the figure. Node color matches with the four grooming stages described in Fig 4. Solid lines represent transitions common with the ND technique. Dashed lines show transitions not found in the ND technique. Transitions with probabilities higher than 90% are indicated with thicker lines. Double-bordered movements (3) are performed only by the disabled individuals that use this technique.

complex of the grooming techniques since it comprises only four movements joined by eight transition lines of which none is common with the main nondisabled grooming technique ND. This technique has a significantly lower movement efficiency than the ND (Mann-Whitney: W = 119, p-value = 0.049) technique. However, no significant difference was found with the NDM technique (Mann-Whitney: W = 144, p-value = 0.094). Moreover, this disabled technique has a lower removal efficiency than ND (Mann-Whitney: W = 27, p-value < 0.001) and NDM (Mann-Whitney: W = 23, p-value < 0.001) (Table C in S4 File).

## Discussion

We found that manual disability in the form of congenital malformation of the digits and forelimbs has a significant effect on grooming efficiency in adult female Japanese macaques at Awajishima. As a group, disabled females were significantly less efficient than their nondisabled conspecifics (removal efficiency 71.45%, movement efficiency 42.06%). Moreover, efficiency decreased when the degree of manual disability increased, revealing that manual physical impairment impacts the ability to remove louse eggs, and that these costs were higher with more extensive physical impairments. These results correspond well with results from studies on manual disability in chimpanzees, where the most severely snare-injured individuals showed reduced feeding efficiency [5].

The analysis of grooming techniques also demonstrated that all females, independent of their physical condition, groom following the same ordered sequence of stages. This finding is similar to results published by Stokes and Byrne (2001) [5], where all chimpanzees used a hierarchically structured leaf processing behaviour, and only injured individuals developed alternative procedures or techniques for accomplishing each one of the tasks involved on this process. However, among the female Japanese macaques in this study, alternative procedures were observed in both disabled and nondisabled groups. For example, nondisabled grooming techniques (ND and NDM) differed in the use of the mouth to remove louse eggs. Although this movement was used more frequently by the disabled females, it constituted a variation in grooming behaviour, where individuals skipped the *Grip the egg* stage and directly used the mouth to eliminate ectoparasites. Such variation was also observed by Tanaka (1995) [27] where samples in which the use of the mouth was observed were not included in the analysis. Furthermore, this use of the mouth behaviour also occurred among some captive nondisabled adult female Japanese macaque at the Granby Zoo in Quebec, Canada, in 2019 (JPEC, personal observation).

Even though both disabled and nondisabled females used the *Mouth directly* (md) gesture, disabled females used it significantly more frequently. This result corroborates the findings of Turner et al. (2012) [6], where nondisabled females rarely used their mouths during grooming while disabled individuals were found to use the mouth more frequently. In our dataset, all disabled females used the mouth while grooming; in contrast, only six nondisabled monkeys were observed to include it on their grooming bouts. The extensive mouth use among disabled monkeys could be explained by the presence of manual physical impairment as well as the reduced manual dexterity experienced by disabled individuals. However, it still remains unclear as to why nondisabled individuals use the mouth while grooming, since they have full ability to do a finger-and-thumb pinch. Ryu et al (2016) found that age-related visual impairment influences grooming [37], and one possibility could be that the presence of visual impairment in older females encouraged them to use their mouth while grooming, however as the 6 nondisabled females who used the *Mouth directly (md)* movement were of different ages, other explanations seem more likely in this case.

Another possibility is that the presence of so many disabled individuals in the population influenced the grooming techniques of the nondisabled females. This could be particularly the case for nondisabled females with close disabled female kin. For instance, having a disabled mother who used her mouth while grooming could have an effect on the daughter's technique. However, we had detailed kin information for only 3 of the 6 nondisabled users of the *Mouth directly (md)* movement: of those, two had disabled mothers. Also, while we found no significant effect of matriline as a predictor variable for mouth use (or number of movements, removal efficiency or movement efficiency), interestingly all six of the nondisabled monkeys who used their mouths in grooming had disabled females in their matrilines, and only one of the nondisabled females who did not use her mouth was known to have a disabled female in her matriline. However, as mouth use in grooming among nondisabled females is not unique to the AMC, and our data on kinship were incomplete, more research would be required to explore this question in depth.

Interestingly, when measuring grooming efficiency as movement efficiency, the nondisabled technique that includes the mouth (NDM), was less efficient than the one without the mouth (ND). This finding suggests that using the mouth might not be as precise at removing ectoparasites as doing a finger-and-thumb pinch. This suggestion is supported by the observation that individuals using the mouth to remove louse eggs often made several attempts before actually removing an egg. In contrast, both techniques show similar removal efficiency, suggesting that having manual dexterity and a complete ability to do a pinch compensated for the additional movements needed when using the mouth for nondisabled monkeys. A nondisabled individual could initially try to remove an egg with the mouth and, if unsuccessful, she could continue the attempt by using her index and thumb fingers to successfully detach it. It is worth mentioning that, in the nondisabled group, the two individuals that removed the most eggs during grooming did not include the mouth in their grooming repertoires, while the two individuals that removed the fewest eggs did. Further research on nondisabled females would be necessary to determine the prevalence and effectiveness of these two grooming techniques among Japanese macaques.

The comparisons made among grooming techniques highlighted how disability leads to the generation of alternative movement sequences and/or modified movements, such as the *Hand/arm sweep* (hs), which was used only by the individuals with highest manual disability index (Ribbon (0.85) and Yuki (0.76)). Moreover, disabled individuals developed novel movements to compensate for their manual impairments. The *Two-hand pinch*, *Two-hand to mouth*, and *Elbow push* movements were observed in individuals who lack fingers on one or both of their hands and used the DF, DE, DC, and DB grooming techniques. During the video analysis, none of the nondisabled females was observed using the elbow for parting the hair, nor both hands together for conducting a pinch or carrying the egg to the mouth. These gestures, used while grooming, were unique to disabled individuals. This observation is consistent with Turner et al.'s research [6,14,15], which described these gestures as relatively rare and used only by disabled females who lacked functional opposability in both hands.

The evidence of new grooming movements by disabled individuals at the AMC is an example of behavioural flexibility describing innovation and problem-solving skills [1]. Macaque females with lower index of manual disability scores, such as Yokam (0.18) who used the grooming technique DA, utilized the malformed hand to do the *Hand push* (hp) gesture for parting the hair and the non-malformed hand to grip the egg using a pinch. This reflects how an individual adapts learned movements to her needs. Similarly, this concept was also applied to snare-injured gorillas and chimpanzees who, after acquiring manual disability, adapted their feeding techniques to their new condition. Furthermore, the degree at which these techniques were adapted by gorillas and chimpanzees was correlated with the degree of manual

disability of the individual [4,5]. This finding provides additional support to the social transmission of grooming behaviours, as an example of imitative learning: they learned how to groom by observation and trial and error, and then adapted the movements according to their impairments [5,27].

The next question to be discussed is if the use of adapted and novel movements compensated for physical impairments in terms of grooming efficiency, measured as removal efficiency (number of eggs removed per 2-minute sample) and movement efficiency (number of movements performed per egg removed). It is interesting to note that the techniques used by individuals with a lower degree of impairment (DA, and DB) were as efficient as the nondisabled technique that includes the mouth (NDM). Compared to the nondisabled technique ND, these techniques were significantly less efficient (DA 17.73%, DB 44.99%) when measured as movement efficiency; however, they were not different in terms of just the removal efficiency (Table C in S4 File). Therefore, even if these disabled females had not adapted or created novel movements, their relatively unaffected manual dexterity enabled them to succeed in the egg-removal task, but since they still had a certain degree of manual disability, it took more movements to remove each egg. Substantial differences in efficiency were found for individuals with medium to high index of manual disability scores. The techniques DF, DE, DD, and DC were not as efficient as the nondisabled grooming techniques (ND and NDM). As a consequence, when groomed by one of these disabled individuals, the grooming recipient would retain a higher louse burden because fewer louse eggs would be removed per grooming bout. In these cases, it could be valid to say that as far as the hygienic function of grooming is concerned, the adapted and novel movements used by disabled females did not compensate for their physical impairment.

A noticeable difference is evident in the results concerning the grooming technique used by Ribbon (DF), the disabled female with the highest manual disability index (0.85). We found no significant difference in the removal efficiency between the DF technique and the nondisabled technique including the mouth NDM. This result could be explained by this subject's sample size used in the study (5 removed eggs in total), given how the statistic was calculated based on the means, and the sample's length (2-minutes). Although this subject had 11 video samples, in many of them no egg was removed, due to the use of the tongue or because she was observed removing debris or dirt particles from the groomee's hair instead of eggs.

We also compared the overall number of movements done in 2 minutes between disabled and nondisabled individuals, and no significant difference was found. Assuming that this lack of difference is not caused by the data structure or the sample size, it could reflect a real similarity in behaviour. This result suggests that the experience of being groomed by a disabled monkey is likely very similar to the experience of being groomed by a nondisabled one, in terms of the frequency of touch and contact. This similarity suggests that disabled monkeys are able to fulfill the social function of grooming, even though they are less efficient in terms of the hygienic function. This illustrates the real importance of the social function of grooming, and how the development of alternate and novel movements seems to be socially useful [6]. In previously published research, Turner et al. (2014) found that disabled females in this population spent more time engaged in rest and less time engaged in social activities compared with nondisabled females [14]. Disabled females had fewer grooming partners compared with nondisabled females. It is possible that the difference in the number of grooming partners reflected a disability-associated bias (partially mitigated by more grooming with kin [14]) due to differences in grooming ability or efficiency, however it seems more likely that it was related to a broader pattern of trade-offs made by the disabled monkeys themselves. Number of grooming partners was one of many social variables in an overall pattern that suggested that disabled females were trading off a reduction in their social time in order to increase their rest time. In

this vein, there was no significant difference in the amount of time disabled females spent being groomed by others. So although disabled females had fewer grooming partners, those partners groomed them for substantial amounts of time, and in most respects disabled females were not differentially treated by their nondisabled conspecifics [6,14]. Also, notably, there was no effect of disability on the proportion of successful grooming solicitations; disabled females were just as likely as nondisabled females to receive grooming after presenting themselves for grooming [14]. Therefore, in the AMC population, disabled individuals seemed to trade grooming based somewhat on the reciprocity principle and behave as though they remember past grooming interactions with other individuals [6,21,38]. Additionally, since the number of movements per sample is similar for disabled and nondisabled groomers in our study, the relaxing function of grooming is likely also fulfilled, with grooming recipients receiving similar social and relaxing benefits from being groomed by a disabled or nondisabled individual. This suggests that, in the biological market, the grooming techniques used by disabled individuals may fulfill the social and relaxing functions of grooming, even if they may carry a hygienic cost for the groomee.

Although studies on provision-fed populations, such as the one of the AMC, provide insights about the effects of physical impairment on an individual's behaviour and fitness proxies, more research is needed to understand how disability affects primates in the wild. At the AMC, there is high food availability and little danger of predation, giving the opportunity for disabled individuals to compensate for physical impairments, to display behavioural flexibility, and to survive and successfully participate in the social activities of the group [14]. In contrast, non-provisioned animals may experience harsher environments and higher competition that could influence their behaviour and reproductive success. These findings add to a growing body of literature on behavioural ecology and the study of behavioural flexibility. This study shows that at least under relatively favourable conditions disabled individuals are able to develop alternative or new behaviours to compensate for their impairments which allowed them to participate in social grooming. However, there may be limits to their ability to compensate for all aspects of their impairment, as shown by the negative impact of manual impairment on the hygienic function of grooming.

## Supporting information

**S1 File. Focal animal names and codes, grooming behavioural ethogram, behaviour/movement images, generalized mixed-effects models run in this study.**
(PDF)

**S2 File. Full statistical results for grooming efficiency and use of mouth for disabled and nondisabled subjects.**
(PDF)

**S3 File. Joint frequencies matrix, transition frequencies matrix, and flowcharts for each grooming technique.**
(PDF)

**S4 File. Statistical results for grooming efficiency of disabled and nondisabled grooming techniques.**
(PDF)

**S1 Dataset. All data underlying the findings reported.**
(CSV)

## Acknowledgments

JPEC would like to express deep gratitude to Willy Montes, for his support and help in the creation of the grooming technique flowcharts. SET would like to thank the Nobuhara family, especially Mr. and Mrs. Toshikazu and Hisami Nobuhara, for permission to conduct research at the Awajishima Monkey Center and their ongoing research support, and Mr. and Mrs. Kosaku and Keiko Okada and family for all the logistical support and kindness; Dr. M. Naka-michi, Dr. K. Shimizu, Dr. K. Yamada, Mr. Y. Kaigaishi, Dr. K. Onishi, Ms. H. Onishi, Dr. A.J. J. MacIntosh, Dr. M. Huffman, Ms. M. Joyce, and Ms. B. Stewart for research support in Japan; Dr. J. Addicott, Dr. M. Pavelka, Dr. S.M. Reader for discussions and advice on research and data-management; and Dr. H.D. Matthews, Dr. K.L. Turner and Dr. N.J. Turner for field assistance in 2007, and Robert Turner for camera equipment advice and sponsorship. We are very grateful to Robert Moriarity for his advice regarding the statistical analysis and to Dr. Jochen Jaeger for his useful comments and suggestions. Finally, all the authors would like to especially thank Dr. Lori K. Sheeran and one anonymous reviewer for their helpful comments and suggestions on this manuscript.

## Author Contributions

**Conceptualization:** Jenny Paola Espitia-Contreras, Sarah E. Turner.

**Data curation:** Jenny Paola Espitia-Contreras, Sarah E. Turner.

**Formal analysis:** Jenny Paola Espitia-Contreras, Sarah E. Turner.

**Funding acquisition:** Sarah E. Turner.

**Investigation:** Jenny Paola Espitia-Contreras.

**Methodology:** Jenny Paola Espitia-Contreras, Sarah E. Turner.

**Resources:** Sarah E. Turner.

**Software:** Jenny Paola Espitia-Contreras.

**Supervision:** Linda M. Fedigan, Sarah E. Turner.

**Validation:** Sarah E. Turner.

**Visualization:** Jenny Paola Espitia-Contreras.

**Writing – original draft:** Jenny Paola Espitia-Contreras.

**Writing – review & editing:** Jenny Paola Espitia-Contreras, Linda M. Fedigan, Sarah E. Turner.

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
