## [Decision Letter · Decision Letter 0]

9 Dec 2019

PONE-D-19-28479

Social grooming efficiency and techniques are influenced by manual impairment in free-ranging Japanese macaques (Macaca fuscata)

PLOS ONE

Dear Mrs Espitia Contreras,

Thank you for submitting your manuscript to PLOS ONE. After careful consideration, we feel that it has merit but does not fully meet PLOS ONE’s publication criteria as it currently stands. Therefore, we invite you to submit a revised version of the manuscript that addresses the points raised during the review process.

We would appreciate receiving your revised manuscript by Jan 23 2020 11:59PM. To enhance the reproducibility of your results, we recommend that if applicable you deposit your laboratory protocols in protocols.io, where a protocol can be assigned its own identifier (DOI) such that it can be cited independently in the future. For instructions see: http://journals.plos.org/plosone/s/submission-guidelines#loc-laboratory-protocols

We look forward to receiving your revised manuscript.

Kind regards,

Bi-Song Yue, Ph.D

Academic Editor

PLOS ONE

**When submitting your revision, we need you to address these additional requirements:**

**Please ensure that your manuscript meets PLOS ONE's style requirements, including those for file naming. The PLOS ONE style templates can be found at http://www.plosone.org/attachments/PLOSOne_formatting_sample_main_body.pdf and http://www.plosone.org/attachments/PLOSOne_formatting_sample_title_authors_affiliations.pdf**

Reviewers' comments:

Reviewer's Responses to Questions

**Comments to the Author**

1. Is the manuscript technically sound, and do the data support the conclusions?

Reviewer #1: Yes

Reviewer #2: Yes

2. Has the statistical analysis been performed appropriately and rigorously? 

Reviewer #1: Yes

Reviewer #2: Yes

3. Have the authors made all data underlying the findings in their manuscript fully available?

Reviewer #1: Yes

Reviewer #2: Yes

4. Is the manuscript presented in an intelligible fashion and written in standard English?

Reviewer #1: Yes

Reviewer #2: Yes

5. Review Comments to the Author

Reviewer #1: I enjoyed reading this paper and appreciated the authors clarity with respect to overall organization, writing, and analysis. The authors provide an interesting contribution to the literature that focuses on behavioral flexibility and primates’ adaptation to disability. I suggest it be accepted for publication in Plos One. I proposed below a few recommendations for the authors to consider in any revision that may occur.

Minor editorial suggestions:

• Line 41 should the world “in” be “any”?

• Lines 52 and 70 comma needed after e.g.? (you used commas after i.e.)

• Line 80 macaque (lower case)

• Line 152 order; however,

• Line 163 samples (lower case)

• Some inconsistencies in case for your ethogram behaviors. It appears that you are following the pattern of capitalizing the first word and lower case for the rest. With that in mind, please see lines 288, 316-319, 331, 341, 375, 388, 398, 399, 402, 415, 455, 487, 488, 498

• Line 513 efficiency; however,

• Line 546 I suggest you delete the word “properly”

• Line 562 conditions (plural)

• Lines 561-566 The information might read more clearly broken into two sentences, for example, at line 564 grooming. However,

Analytical suggestions:

N.B.: In my comments below, D=Disabled individuals, ND=Nondisabled individuals

• Lines 542-544 some additional discussion of this might be helpful. It occurs to me that one response that a potential groomee might have to being approached for grooming by a D (and therefore less efficient) individual would be to refuse the invitation to groom. The source cited on lines 544-5 (14) may address this, but it was unclear to me whether you meant D individuals being invited to groom others, or D individuals being groomed. I am wondering if D individuals groom more frequently with relatives (see also below). I wondered whether D females had fewer grooming partners compared to ND females…I apologize if you reported this, and I missed it! But I couldn’t easily find it on my first or second read. I suspect that is the case and thought it might relate to your discussion of reduced efficiency by ND females.

• An interesting characteristic of this population is that you have information on maternal relatedness. As I read your paper, I wondered how matrilineal relationships might impact on your data and/or on your interpretations of it. Lines 82 -87 raise the issue of technique being transmitted through kin lines. Are you arguing in the following sentences that trial and error learning overrides this? I felt like this could be discussed more thoroughly at lines 464-467 and specifically wondered whether your six ND females who used the mouth directly to groom were related to any of the D females. It would be helpful to indicate kin structure in S1 table, as I (again) wondered about the relationships among the females you studied. Lines 440-454 is another point in the manuscript where I had questions about kinship and how matriline membership might be (or might not be!) patterning the data. If possible, I think your arguments would be strengthened by including matriline as a variable in your analysis. It might also be helpful to indicate whether disabilities in this population are clustered within matrilines, or are disabilities distributed across the population?

• Another compelling feature of your population is that you likely have more individuals living to be old than is true of unprovisioned populations. I did not see age reported in S1 table, and I suggest that be included. The disability classifications you used in this paper focused on those that influence manual dexterity/grip, but it occurred to me that visual impairment might be occurring in your older females (similar to that reported for bonobos by Ryu, Graham, Sakamaki, and Furuichi Current Biology 26, R1119–R1136). As was true for matriline membership, I wondered whether age explained any of the patterning in your dataset. If your ND mouth directly groomers are old females, that might account for their use of that technique. It seems that relying more on one’s mouth might occur as the ability to see the louse is reduced or lost.

Thank you for this opportunity to review your interesting work.

Reviewer #2: This manuscript presents a novel analysis of grooming technique in disabled Japanese macaques, taking advantage of a population with a high degree of disability to accomplish this. Theoretical background is multi-faceted and well researched. Methods are clear and well-suited to address their hypotheses. The results are presented clearly. Discussion covers needed areas well.

6. PLOS authors have the option to publish the peer review history of their article (what does this mean?). If published, this will include your full peer review and any attached files.

Reviewer #1: Yes: Lori K. Sheeran

Reviewer #2: No

---

## [Author Response · Author response to Decision Letter 0]

21 Jan 2020

Reviewer #1:

I enjoyed reading this paper and appreciated the authors clarity with respect to overall organization, writing, and analysis. The authors provide an interesting contribution to the literature that focuses on behavioral flexibility and primates’ adaptation to disability. I suggest it be accepted for publication in Plos One. I proposed below a few recommendations for the authors to consider in any revision that may occur.

Thank you for the positive assessment of our research and for taking the time to provide us with your valuable suggestions, thoughts and feedback.

Minor editorial suggestions:

• Line 41 should the world “in” be “any”?

Done (Line 41)

• Lines 52 and 70 comma needed after e.g.? (you used commas after i.e.)

Done (Line 52 and 70)

• Line 80 macaque (lower case)

Done (Line 80)

• Line 152 order; however,

Done (Line 153)

• Line 163 samples (lower case)

Done (Line 164)

• Some inconsistencies in case for your ethogram behaviors. It appears that you are following the pattern of capitalizing the first word and lower case for the rest. With that in mind, please see lines 288, 316-319, 331, 341, 375, 388, 398, 399, 402, 415, 455, 487, 488, 498

Done. Changes are located at lines 267, 268, 291, 319-322, 334, 378, 391, 401, 402, 405, 418, 458, 504, 505, 515. We applied this suggestion to all the behaviors listed in the ethogram in both the manuscript and the supplementary material S1.

• Line 513 efficiency; however,

Done (Line 530)

• Line 546 I suggest you delete the word “properly”

Done (Line 574)

• Line 562 conditions (plural)

Done (Line 591)

• Lines 561-566 The information might read more clearly broken into two sentences, for example, at line 564 grooming. However, 

Done (Line 593)

Analytical suggestions:

N.B.: In my comments below, D=Disabled individuals, ND=Nondisabled individuals

Lines 542-544 some additional discussion of this might be helpful. It occurs to me that one response that a potential groomee might have to being approached for grooming by a D (and therefore less efficient) individual would be to refuse the invitation to groom. The source cited on lines 544-5 (14) may address this, but it was unclear to me whether you meant D individuals being invited to groom others, or D individuals being groomed.

This is an interesting point. Unfortunately, we don’t have the contextual behavioural data to adequately address it here. That is, in the field at AMC, it is generally possible to identify when an animal is “presenting themselves for grooming” (they walk up to another individual and lie down right in front of them, sometimes inserting themselves between others and underneath the other monkey’s hands). We have data on the disability status of those who are involved in this “groom present” behaviour during focal follows and we detected no disability-associated differences (see Lines 569 – 571) for rewritten clarification of this point). What is more difficult to distinguish is if a monkey approaches another monkey with the intention to groom them (that is, not to ask for grooming, but to provide it). We are not able to reliably distinguish between approach-with-the-intention-to-groom versus a generalized approach (except post-hoc, if grooming occurs, which would not reveal the times when grooming was intended but rejected). Also, approach-depart data were not collected in this dataset, so although we might otherwise examine the dynamics of approach-depart in relation to disability, that will have to await future studies (we have started collecting this information as part of focal samples). All this to say, yes, this would be so interesting to know, but at the moment, we don’t have the data to examine this question. However, based on your comments, we have now clarified the information we do have regarding groom-present behaviours and disability, as well as kin relationships in the text (Lines 558-573).

• I am wondering if D individuals groom more frequently with relatives (see also below).

In an earlier study, we found that for all females, available kin was positively associated with the number of grooming partners, however, kin availability had a stronger influence on number of grooming partners for disabled females, compared to nondisabled females. We’ve added a note about this to the new parts of the discussion on kin and grooming (Lines 561 – 562)

• I wondered whether D females had fewer grooming partners compared to ND females…I apologize if you reported this, and I missed it! But I couldn’t easily find it on my first or second read. I suspect that is the case and thought it might relate to your discussion of reduced efficiency by ND females. 

Thanks. We did report this (line 542; now lines 558 - 561) but can see that it needed clarification. We have re-written this information to clarify and highlight it better (please see lines 558 - 569). We have integrated this information with other aspects of your suggestions, including new text in two parts of the discussion (as above, lines 558 - 573, and lines 467 - 482). Data for these social measures come from references 6 and 14. Disabled females in this group had fewer grooming partners but spent a similar amount of time being groomed compared with nondisabled individuals.

• An interesting characteristic of this population is that you have information on maternal relatedness. As I read your paper, I wondered how matrilineal relationships might impact on your data and/or on your interpretations of it. Lines 82 -87 raise the issue of technique being transmitted through kin lines. 

Thanks for your comments. Yes, such a potentially interesting topic! We’re working on this now in another study. We have also added some text to the discussion to better address this possibility (please see above).

• Are you arguing in the following sentences that trial and error learning overrides this? 

Not necessarily, we think it doesn’t override the transmission of techniques by kinship but complements it. In the case of disabled animals, we think that they imitate their relatives’ technique(s) and then develop individually appropriate styles through trial and error. Unfortunately our data on kinship were not detailed enough to allow for an in depth analysis of this topic, but based on your comments, we ran some tests (Kruskal-Wallis tests on various measures of kinship using a coefficient of relationship and on matrilines, however we found no significant results – see below). We are currently investigating the influence of kin and social network on grooming styles in a new study. Although we have some kin relationships in this study, we don’t have complete information: we had only 6 mother-daughter pairs and saw no obvious relationships, but the sample size was too small to draw significant conclusions.

• I felt like this could be discussed more thoroughly at lines 464-467 and specifically wondered whether your six ND females who used the mouth directly to groom were related to any of the D females. 

Thanks, we added to the discussion in lines 472 – 482 to address this idea. 2 of the 6 nondisabled females have a disabled mother, 1 has a nondisabled mother, and there is no kin information about the other 3, so the sample size was too small to draw many conclusions. 

• It would be helpful to indicate kin structure in S1 table, as I (again) wondered about the relationships among the females you studied. 

Good idea. Thanks! We added a column in Table A in S1 with the maternal kinship information we had available, and a separate column for matriline.

• Lines 440-454 is another point in the manuscript where I had questions about kinship and how matriline membership might be (or might not be!) patterning the data. If possible, I think your arguments would be strengthened by including matriline as a variable in your analysis. It might also be helpful to indicate whether disabilities in this population are clustered within matrilines, or are disabilities distributed across the population?

When we were doing our initial analyses for this paper, we had put aside the idea of examining kinship because our kinship dataset is incomplete in various ways, and there are some females for whom kinship data are lacking. However, you raised some very good points, and so we explored this question further. We conducted a series of Kruskal-Wallis tests on some different metrics of kin availability (using mother, daughter, grandmother, granddaughter coefficients of relationship and mother, daughter, aunt, niece, grandmother, granddaughter coefficients of relationship also). Then, separately, we conducted Kruskal-Wallis tests with matriline as a predictor variable. We also ran some linear models to see if there were any significant interactions occurring.

We found no significant effect of matriline as a predictor of mouth use (or number of movements, removal efficiency, or movement efficiency). However, we noted that all six of the nondisabled monkeys who used their mouths during grooming had disabled monkeys in their matrilines, only one nondisabled female who was known to have a disabled female in her matriline did not use her mouth while grooming. We have now added text to the discussion to highlight this information and clarify matriline and age as potential variables in the study (Lines: 196-199 and, as above 467 - 482)

The possibility that disability-associated grooming styles are transmitted within matrilines is interesting and we are currently investigating this question in the field, along with questions about social grooming network.

• Another compelling feature of your population is that you likely have more individuals living to be old than is true of unprovisioned populations. I did not see age reported in S1 table, and I suggest that be included. 

Yes, good point. We have now added age to Table A in S1.

• The disability classifications you used in this paper focused on those that influence manual dexterity/grip, but it occurred to me that visual impairment might be occurring in your older females (similar to that reported for bonobos by Ryu, Graham, Sakamaki, and Furuichi Current Biology 26, R1119–R1136). As was true for matriline membership, I wondered whether age explained any of the patterning in your dataset. If your ND mouth directly groomers are old females, that might account for their use of that technique. It seems that relying more on one’s mouth might occur as the ability to see the louse is reduced or lost.

Thanks for this reference and suggestion. To find if Age had a significant effect on grooming efficiency, we conducted again the statistical tests including Age as fixed factor and all of them resulted in nonsignificant outcomes. In addition to that, we noted the use of Mouth directly (md) was observed in ND females of different ages (not only in old ones). The visual impairment hypothesis is an interesting idea, but our limited data do not support it at this point. However, this idea was added to the discussion (lines 467 - 471) for future work, and in the section on mouth use in grooming (see above).

Thank you for this opportunity to review your interesting work.

Dr. Sheeran, thank you so much for the comments, your thoughtful suggestions and encouragement, we hope that you will find the manuscript substantially improved. We appreciate your thoughts and feedback!

Reviewer #2:

This manuscript presents a novel analysis of grooming technique in disabled Japanese macaques, taking advantage of a population with a high degree of disability to accomplish this. Theoretical background is multi-faceted and well researched. Methods are clear and well-suited to address their hypotheses. The results are presented clearly. Discussion covers needed areas well.

Thank you very much for taking the time to read our manuscript and for the positive assessment of our research.

---

## [Editor Report · Decision Letter 1]

29 Jan 2020

Social grooming efficiency and techniques are influenced by manual impairment in free-ranging Japanese macaques (Macaca fuscata)

PONE-D-19-28479R1

Dear Dr. Espitia Contreras,

We are pleased to inform you that your manuscript has been judged scientifically suitable for publication and will be formally accepted for publication once it complies with all outstanding technical requirements.

With kind regards,

Bi-Song Yue, Ph.D

Academic Editor

PLOS ONE

---

## [Editor Report · Acceptance letter]

10 Feb 2020

PONE-D-19-28479R1 

Social grooming efficiency and techniques are influenced by manual impairment in free-ranging Japanese macaques (*Macaca fuscata*) 

Dear Dr. Espitia-Contreras:

I am pleased to inform you that your manuscript has been deemed suitable for publication in PLOS ONE. Congratulations! Your manuscript is now with our production department. 

With kind regards,

on behalf of

Dr. Bi-Song Yue 

Academic Editor

PLOS ONE